# Patterns of Signal Intensity in CISS MRI of the Inner Ear and Eye

Antonia Mair [1], Christopher I. Song [1], Bela Büki [2] and Bryan K. Ward [1,*]

1  Department of Otolaryngology-Head and Neck Surgery, The Johns Hopkins University School of Medicine, Baltimore, MD 21205, USA; antonia.mair@yahoo.de (A.M.); csong29@alumni.jh.edu (C.I.S.)
2  Department of Otolaryngology, Karl Landsteiner University Hospital Krems, Mitterweg 10, A-3500 Krems an der Donau, Austria
*  Correspondence: bward15@jh.edu

**Abstract:** Background: Constructive interference in steady state (CISS) is a gradient echo magnetic resonance imaging (MRI) pulse sequence that provides excellent contrast between cerebrospinal fluid and adjacent structures but is prone to banding artifacts due to magnetic field inhomogeneities. We aimed to characterize artifacts in the inner ear and eye. Methods: In 30 patients (60 ears/eyes) undergoing CISS sequence MRI, nine low-signal intensity regions were identified in the inner ear and compared to temporal bone histopathology. The number and angle of bands across the eye were examined. Results: In the cochlea, all ears had regions of low signal corresponding to anatomy (modiolus (all), spiral lamina ($n = 59$, 98.3%), and interscalar septa ($n = 50$, 83.3%)). In the labyrinth, the lateral semicircular canal crista ($n = 42$, 70%) and utricular macula ($n = 47$, 78.3%) were seen. Areas of low signal in the vestibule seen in all ears may represent the walls of the membranous utricle. Zero to three banding artifacts were seen in both eyes (right: 96.7%, mean 1.5; left: 93.3%, mean 1.3). Conclusion: Low signal regions in the inner ear on CISS sequences are common and have consistent patterns; most in the inner ear represent anatomy, appearing blurred due to partial volume averaging. Banding artifacts in the eye are more variable.

**Keywords:** banding artifacts; MRI; CISS sequences; eye; inner ear

## 1. Introduction

Magnetic resonance imaging (MRI) is a tool quickly advancing in clinical evaluations of skull base and temporal bone pathologies, including those affecting the inner ear. Heavily T2-weighted sequences, such as constructive interference in steady state (CISS), are frequently used to image the skull base due to the excellent contrast between cerebrospinal fluid (CSF) and surrounding structures such as cranial nerves. These sequences are regularly used for assessing cholesteatoma and asymmetric sensorineural hearing loss. They are reliable tools for imaging tumors of the cerebellopontine angle, evaluating inflammatory or infectious processes of the inner ear, and determining the patency of the cochlea to receive a cochlear implant [1]. Heavily T2-weighted images can be generated using either gradient echo or spin echo pulse sequences. Both approaches are used in clinical practice but have different advantages and disadvantages. T2*-weighted gradient echo sequences such as CISS can be obtained more rapidly, but they are prone to banding artifacts—linear, low signal intensity stripes—that traverse anatomic structures [1,2]. T2-weighted spin echo sequences, such as sampling perfection with application-optimized contrast using different flip angle evolution (SPACE), have longer image acquisition times and are more prone to artifacts from motion but are less sensitive than gradient echo sequences to inhomogeneities in the magnetic field. In SPACE images of the inner ear, some areas of low signal within the inner ear correlate to anatomical structures within the vestibule, including the lateral semicircular canal crista and the utricular macula. In contrast, others do not, potentially representing artifacts of fluid movement within the endolymphatic space caused by magneto-hydrodynamic Lorentz forces [3]. A similar study of the inner ear to distinguish

anatomy from artifacts for gradient echo sequences has yet to be performed and is one of the aims of this study.

Banding artifacts on T2*-weighted sequences are also commonly seen in the eye. The rise in clinical referrals for MRI of the orbit for common conditions like cataracts, myopia, glaucoma, and presbyopia has increased the importance of assessing areas of signal changes, such as artifacts that could complicate diagnostic assessments [4,5]. Previous studies on MR artifacts affecting the eye have focused mainly on the optic nerve, anterior and posterior chamber, ciliary body, iris, lens, and extraocular muscles but have not examined banding artifacts within the globe [6]. It has been shown that artifacts in the inferior orbital region can be mistaken for neoplastic or inflammatory orbital diseases, potentially leading to unnecessary orbital biopsies or misdiagnoses [7]. To analyze if banding artifacts of the eye are potentially homogenous and thus resulting from the magnetic field or are part of biological signals underlying pathophysiological processes in diseases of the eye, the angles of the bands were analyzed [8].

Accurately distinguishing between artifacts and anatomical or pathological structures is essential in MRI interpretation to ensure an accurate diagnosis and prevent unnecessary further investigations. This study aimed to analyze banding artifacts on axial MRI CISS sequences of the inner ear and eye, addressing a critical gap in the literature on interpreting MRI of the inner ear.

## 2. Materials and Methods

### 2.1. Research Design and Patient Selection

Patients who underwent an MRI of the skull base at the Johns Hopkins Hospital between 1 April 2017, and 1 November 2019 were screened for axial CISS sequences. A query was performed for MRI scans of the brain with and without intravenous contrast in one location in the adult hospital. The query returned 7290 scans. The scans were then filtered for those performed using the same 3 Tesla scanner (Siemens Magnetom Skyra, Forchheim, Germany, *n* = 1521 scans). Scans were excluded if the patient was under 18 years of age and if the images did not fully include the eyes and inner ears. Additionally, patients were screened for cerebellopontine angle tumors and diseases of the eye potentially leading to magnetic field disturbances. Patients with cerebellopontine angle or inner ear tumors, ocular pathologies secondary to cranial pathologies (e.g., pituitary adenoma causing visual field defects), or systemic diseases, such as diabetes mellitus causing diabetic retinopathy, were excluded. The first 30 consecutive patients who underwent a skull-base protocol that included higher resolution (0.6 mm isometric voxels) CISS sequences and who met the above criteria were selected for further analysis. A flow chart for the included data is shown in Figure 1. This retrospective study was approved by the Institutional Review Board of Johns Hopkins University School of Medicine as project number IRB00279939. Informed consent was waived due to the retrospective nature of this study.

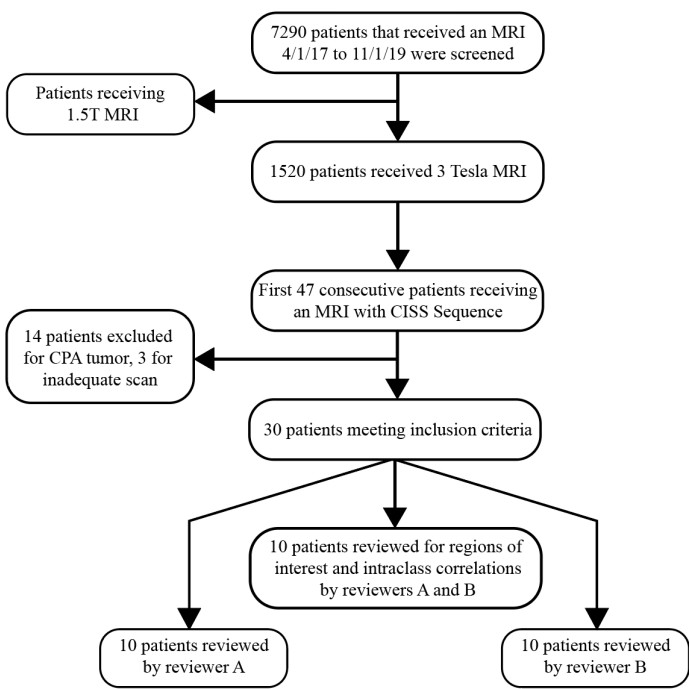

**Figure 1.** Patient Selection and Data Inclusion Flowchart. This diagram outlines the sequential screening and selection process for patients from the initial MRI screening to the final review.

## 2.2. Image Acquisition and Post-Processing

Pulse sequence parameters for the included scans are as follows: TR 5.46 ms (range 5.45–6.26 ms); TE 2.43 ms (range 2.42–2.75 ms); flip angle 42.00° (range 40.00–42.00°); SW 0.60 mm. Three investigators (A.M., C.S., and B.W.) reviewed the first ten MRI scans to identify areas of low signal intensity within the inner ear. After initial inspection, nine regions of the inner ear were identified in consensus based on anatomical landmarks and then systematically assessed in all scans by A.M. and C.S. To determine whether the areas of low signal intensity on MRI corresponded to anatomic structures, regions of low signal intensity were compared to axial hematoxylin and eosin-stained sections of the inner ear from an adult without a history of inner ear disease. CISS images of the eye were reviewed for banding artifacts using Carestream Vue Motion and then processed using Phillips Vue PACS (Carestream Health, Inc., Rochester, NY, USA). The number of banding artifacts in each eye and the angle of these bands were measured using tools within the software. Two angles were measured: the angle of the limbs of the band through the eye using the center of the lens to mark the vertex (i.e., the internal angle), and the angle between the band's lateral aspect and a plane drawn between both optic nerves (Figure 2). The average of the measured angles was recorded if multiple banding artifacts were present in the eye.

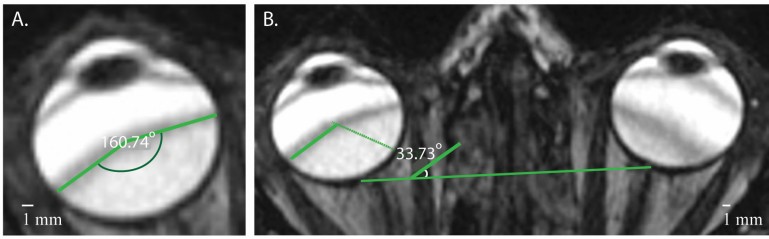

**Figure 2.** Measurement technique for the angle of the bands. Panel (**A**) illustrates how the vertex is marked using the lens as a reference point, and then an angle is measured between the two edges (limbs) of the band. Panel (**B**) shows the measurement of the angle at which the outer edge (limb) of the band forms with a reference plane that is established by drawing a line connecting the two optic nerves.

*2.3. Statistical Analyses*

Following consensus regarding the areas of low signal intensity, the two readers each assessed the same ten scans (20 ears). Interrater agreement was calculated using Cohen's kappa (K), and the agreement between both investigators was evaluated via the intraclass correlation coefficient (ICC). Each investigator then reviewed half the remaining scans. In total, 30 scans were included, with each having the left and right inner ears and eyes (60 ears and eyes). Comparisons were made between the right and left eyes using a paired *t*-test. The correlation between the number of bands and the presence or absence of eye pathologies was assessed using the Pearson correlation coefficient. Statistical analysis was performed using SPSS (version 29.0, IBM, Armonk, NY, USA). A two-tailed *p*-value < 0.05 was considered indicative of statistical significance.

**3. Results**

*3.1. Signal Areas in the Inner Ear*

Of the 30 patients who were retrospectively identified, 60% (*n* = 18) were female and 40% (*n* = 12) were male, with a median age of 51.1 ± 19.5 years (Table 1). The clinical diagnosis of the 30 patients is shown in Table 1. Nine patterns of low signal intensity were identified, 6 of which were in the vestibular region and 3 in the cochlear region (Figure 3): (A) Arc around the lateral semicircular canal; (B) Indentation in the anterior region of the lateral semicircular canal; (C) Diagonal region across the lateral semicircular canal; (D) Ellipsoid shape in the lateral vestibular region; (E) Pentagon-shaped structure in the center of the vestibulum; (F) "Y"-shaped figure in the posterior vestibular region; (G) Linear stripe along the turn of the cochlea; (H) "V"-shaped region at the top of the cochlea; (I) Region of the center of the cochlea. The first ten scans (20 ears) analyzed by two independent readers yielded an excellent ICC of 0.981 and interrater agreement for identifying areas of low signal using Cohen's kappa (K = 0.962) (Table 2).

**Table 1.** Patient characteristics.

|  |  | Patients (N = 30) |
|---|---|---|
| **Age, Median ± SD** |  | **51.1 ± 19.5 Years** |
| Sex (n) | Male | 12 (40%) |
|  | Female | 18 (60%) |
| Initial diagnosis for imaging | Pituitary adenoma | 9 (30.0%) |
|  | Pituitary tumor | 4 (13.4%) |
|  | Sellar/clival mass | 4 (13.4%) |
|  | Meningioma | 2 (6.7%) |
|  | Craniopharyngioma in the third ventricle | 1 (3.4%) |
|  | Skull base squamous cell carcinoma | 1 (3.4%) |
|  | Right-sided head and face pain | 1 (3.4%) |
|  | 6th nerve paralysis, left eye ptosis, left-sided dysmetria | 1 (3.4%) |
|  | 6th nerve palsy | 1 (3.4%) |
|  | Trigeminal neuralgia | 1 (3.4%) |
|  | Calvarial metastatic disease | 1 (3.4%) |
|  | Pain behind right ear with headache and vertigo | 1 (3.4%) |
|  | Skull base chondrosarcoma | 1 (3.4%) |
|  | Sellar/suprasellar arachnoid cyst | 1 (3.4%) |
|  | Skull base tumor | 1 (3.4%) |

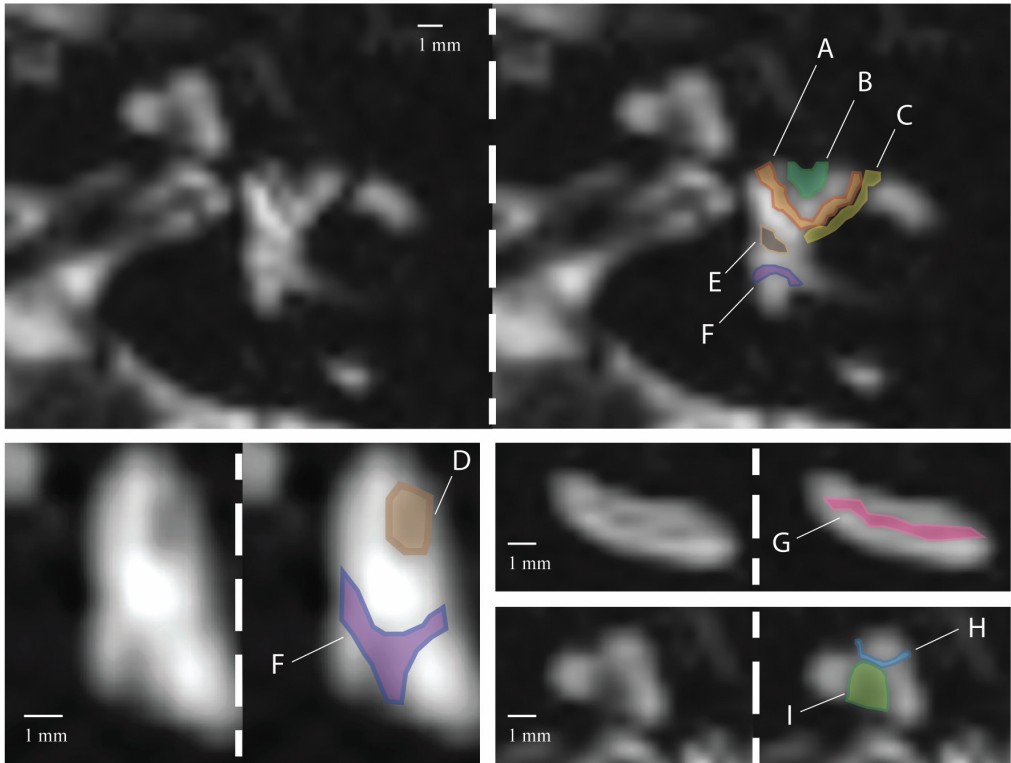

**Figure 3.** Regions of interest in the inner ear. (**A**) Arc around the lateral semicircular canal; (**B**) Indentation in the anterior region of the lateral semicircular canal; (**C**) Diagonal region across the lateral semicircular canal; (**D**) Ellipsoid shape in the lateral vestibular region; (**E**) Pentagon-shaped structure in the center of the vestibulum; (**F**) "Y"-shaped figure in the posterior vestibular region; (**G**) Linear strip along the turn of the cochlea; (**H**) "V"-shaped region at the top of the cochlea; (**I**) Region in the central cochlea.

**Table 2.** Frequency of low signal intensity regions in the vestibule and cochlea sample 10 scans (20 ears) for both readers.

| | | \multicolumn{9}{c}{*Regions of Low Signal Intensity within the Inner Ear (n)*} |
|---|---|---|---|---|---|---|---|---|---|---|
| | | **A** | **B** | **C** | **D** | **E** | **F** | **G** | **H** | **I** |
| **Right ear** | Reader 1 | 8 | 6 | 6 | 7 | 9 | 9 | 8 | 10 | 10 |
| | Reader 2 | 8 | 6 | 5 | 7 | 9 | 9 | 8 | 9 | 10 |
| **Left ear** | Reader 1 | 10 | 7 | 4 | 7 | 10 | 10 | 8 | 10 | 10 |
| | Reader 2 | 10 | 7 | 4 | 7 | 10 | 10 | 8 | 10 | 10 |

Reader 1 (C.S.), Reader 2 (A.M.).

Signal loss in the vestibular region was seen frequently in areas of the "Y"-shaped figure in the posterior vestibule (region F, 96.7%), within the pentagon-shaped structure in the center of the vestibule (region E, 90%), and the arc around the lateral semicircular canal (region A, 91.7%) (Table 3). A diagonal region across the lateral semicircular canal was seen only in 38.5% of ears (C). When compared to the histology, the indentation in the anterior wall of the lateral semicircular canal (region B) corresponded to the cristae of the lateral semicircular canal, and the ellipsoid region in the lateral vestibule corresponded to the utricle (region D). The infrequent diagonal region (C) corresponded to a bony narrowing just lateral to the ampullae of the lateral semicircular canal (Figure 4). The pentagon-shaped region, E, the "Y"-shaped figure in the posterior vestibule, F, and a portion of the arc around the lateral semicircular canal, A, were identified consistently. These areas may represent

portions of the membranous utricle seen in histology. There were no areas within the vestibule that could not be accounted for by inner ear anatomy.

**Table 3.** Frequency of regions of low signal intensity in the vestibule and cochlea.

| | A | B | C | D | E | F | G | H | I |
|---|---|---|---|---|---|---|---|---|---|
| | *Regions of Low Signal Intensity (n)* | | | | | | | | |
| **Right ear** | 27 (90.0%) | 21 (70.0%) | 10 (30.3%) | 22 (73.3%) | 28 (93.3%) | 29 (96.7%) | 24 (80.0%) | 29 (96.7%) | 30 (100%) |
| **Left ear** | 28 (93.3%) | 21 (70.0%) | 14 (46.7%) | 25 (83.3%) | 26 (86.7%) | 29 (96.7%) | 26 (86.7%) | 30 (100%) | 30 (100%) |

(A) Arc around the lateral semicircular canal; (B) Indentation in the anterior region of the lateral semicircular canal; (C) Diagonal region across the lateral semicircular canal; (D) Ellipsoid shape in the lateral vestibular region; (E) Pentagon-shaped structure in the center of the vestibulum; (F) "Y"-shaped figure in the posterior vestibular region; (G) Linear strip along the turn of the cochlea; (H) "V"-shaped region at the top of the cochlea; (I) Region in the central cochlea.

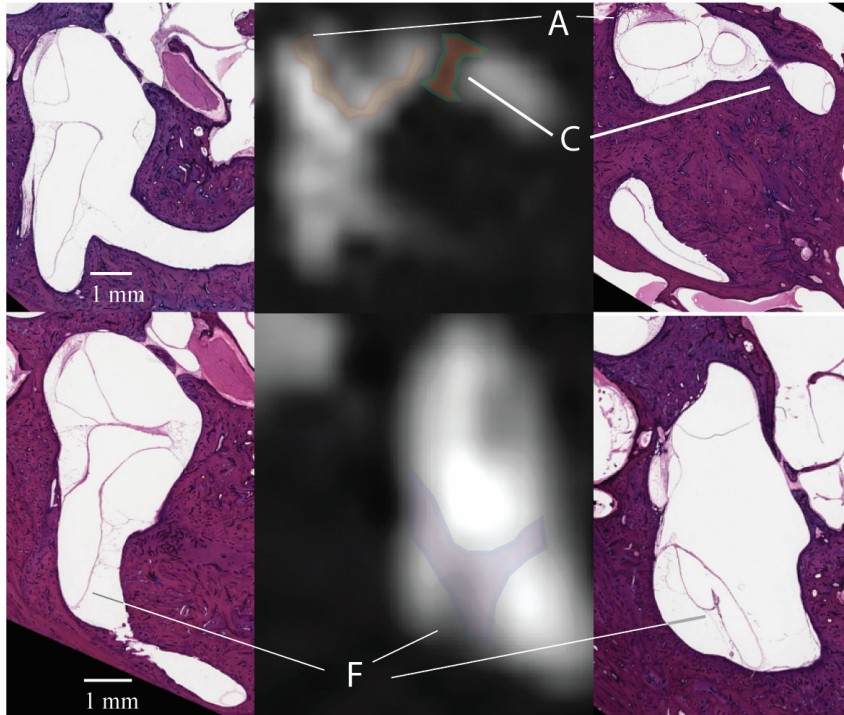

**Figure 4.** Vestibular anatomy may correspond to areas of low signal intensity observed in MRI. Line A highlights a ridge near the lateral semicircular canal, which aligns with the root of the utricle. C identifies the junction of bone just lateral to the ampulla of the lateral semicircular canal and F demonstrates the posterior region in the vestibule. Within the vestibule, the walls of the membranous labyrinth cross the otherwise fluid-filled inner ear, which may be represented as fine, gray lines in MRI scans.

Regions of signal loss in the cochlea were observed in all ears, with 100% of the ears having signal loss in the region of the center of the cochlea, 98.4% in the "V"-shaped region at the top of the cochlea, and 83.4% in a stripe along the turns of the cochlea (Table 3). Compared to the histology, the regions of low signal intensity in the cochlea corresponded to the modiolus, interscalar septa, and osseous spiral lamina, respectively, all representing bony projections within or between turns of the cochlea (Figure 5). None of the low-signal intensity areas within the cochlea were banding artifacts.

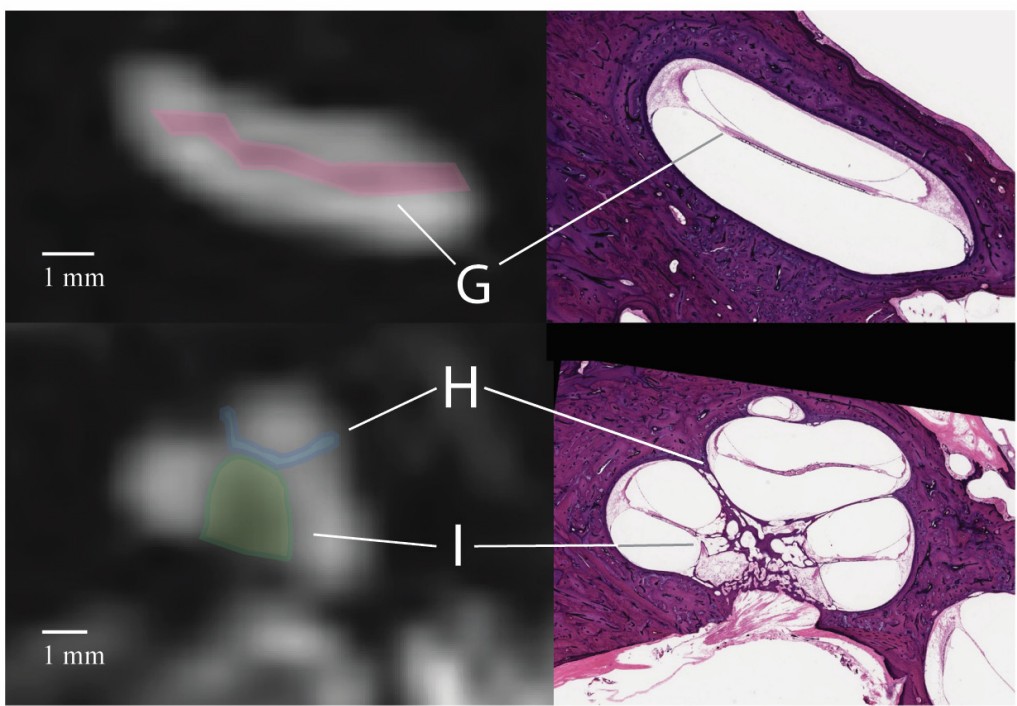

**Figure 5.** Cochlear anatomy corresponds to areas of low signal intensity. G represents the osseous spiral lamina, I the interscalar septum, and H the modiolus. No banding artifacts were seen in the cochlea.

### 3.2. Signal Areas in the Eye

Banding artifacts were identified in 96.7% of the right and 93.3% of the left eyes (two-sample Z test for proportions, $p$ = 0.28). A greater number of bands were identified in the right eye (mean 1.47 (range 0–3) bands in the right eye vs. mean 1.27 (range 0–3) bands in the left eye, paired $t$-test, $p$ = 0.01, Table 4). The mean internal angle of banding artifacts was greater in the left eye compared to the right eye (148.10° left vs. 142.60° right, paired $t$-test, $p$ = 0.04). Eye pathologies were reported in 7 patients (23.3%). They included: open-angle glaucoma, cataracts, optic nerve atrophy, blepharitis of the upper and lower eyelids, strabismus, pseudophakia, optic neuritis, myopia with astigmatism and presbyopia, dystrophy of anterior cornea, corneal dystrophy, blepharoptosis, and cystoid macula edema. Patterns were not observed between the presence of eye pathology and the presence or frequency of bands.

**Table 4.** Number of banding artifacts and mean angle of these artifacts within the band and with the optic nerve plane.

| | *Banding Artifacts of the Eye* | | |
|---|---|---|---|
| | **Mean (SD) Number of Bands in the Eye (n)** | **Mean (SD) Internal Angle of the Banding Artifacts (°)** | **Mean (SD) Angle of Banding Artifacts in Relation to the Optic Nerve Plane (°)** |
| **Right eye** | * 1.47 (1.41) | * 142.60 (13.76) | 58.01 (24.66) |
| **Left eye** | * 1.27 (0.64) | * 148.10 (14.21) | 54.47 (24.46) |

SD, standard deviation, * indicates statistical significance with a paired $t$-test, $p < 0.05$.

### 4. Discussion

Anatomic detail was observed in the inner ear on CISS sequences using 3T MRI. Surprisingly, many of the areas suspected of representing artifacts had likely or potential anatomic correlates on the example histopathology specimen. All linear low-signal areas on the CISS sequence in the cochlea were attributed to anatomic structures, not banding

artifacts. Three areas of decreased signal intensity were described in the cochlea: a linear stripe along the turn of the cochlea, a "V"-shaped region at the top of the cochlea, and a region near the center of the cochlea. The linear region of signal loss between the scala vestibuli and the scala tympani corresponds to the spiral lamina, a bony projection from the modiolus of the cochlea. On axial T2-weighted sequences, this was especially prominent within the basal turn of the cochlea and was present in 83.3% of ears here. In 98.3% of the ears, a V-shaped area of decreased signal at the top of the cochlea was observed. This area corresponds to the interscalar septa, a bony plate that separates the cochlear turns and can form a V-shape towards the cochlear apex. Finally, the signal loss in the central cochlea correlated with the modiolus. All three areas represented bony projections into the fluid-filled inner ear.

In the vestibule, six regions with low signal intensity were observed. The Y-shaped figure in the posterior vestibule was the most frequently seen (96.7%), along with the pentagon-shaped structure in the center of the vestibule (90.0%) and the arc surrounding the lateral semicircular canal (91.7%). Unlike the cochlea, the areas of low-signal intensity in the vestibule do not correlate with bony projections into the fluid-filled space, except for the infrequently seen region C, which we suspect represents bone just lateral to the ampulla of the lateral and superior semicircular canals. The variable frequency of this observation likely reflects differences in the slice angle of each image. The cristae of the lateral semicircular canal also contain bony projections from the labyrinth wall. They can be seen as slight indentations in the walls of the ampulla, likely representing region B.

Several thin lines traverse the vestibule on the CISS sequence MRI. While these thin lines were initially suspected of representing banding artifacts, comparisons to histology revealed their correlation to the walls of the membranous utricle that span the fluid-filled vestibule. Small anatomic structures within the inner ear are smaller than the tiniest 3-dimensional (3-D) box or voxel obtained with current MRI techniques. Therefore, the MRI signal from that structure is inevitably averaged with the nearby endolymph and perilymph and displayed as a gray-scale pixel. This process is shown in Figure 6. The thin walls of the membranous labyrinth may appear as a slightly grayer line of pixels adjacent to the high signal, white inner ear fluid. For bony structures like those spanning the cochlea, the lines are more apparent because of the lower signal of bone on MRI. In other words, the average of a black-and-white object (bone and fluid) will appear as a darker pixel than that of a gray-and-white object (soft tissue and fluid).

Image contrast with adjacent structures is critical to differentiating structures on MRI. The greater the difference in pixel brightness between neighboring pixels, the more likely the object will be resolved. So, structures like the saccular macula are difficult to discern because they sit within the bony spherical recess. Both adjacent structures have low signal intensity and appear dark. However, other structures, like the utricular macula, float freely within the fluid-filled vestibule and are densely packed with cells and otoconia, allowing image contrast. Supporting that this structure can be seen, areas corresponding to the utricular macula have previously been described on spin echo sequences of the inner ear [3,9]. The walls of the membranous labyrinth also traverse the vestibule but may only be visible as a string of a few adjacent pixels due to partial volume averaging. Starting from histopathology, Figure 7 demonstrates how applying signal averaging with decreased spatial resolution can lead to a blurred region of low signal intensity. The example in Figure 7 is a 2-dimensional (2-D) representation of signal averaging, whereas MRI will average the signal from a 3-D tissue volume, further blurring the image. Considering the above, the utricular macula ought to be seen frequently as a distinct structure (object D, upper panel of Figure 7), whereas the thinner structures, like the walls of the membranous labyrinth, would appear as less distinct areas of gray (objects A, E, and F). Scans with smaller voxel sizes are likelier to see these areas as distinct structures.

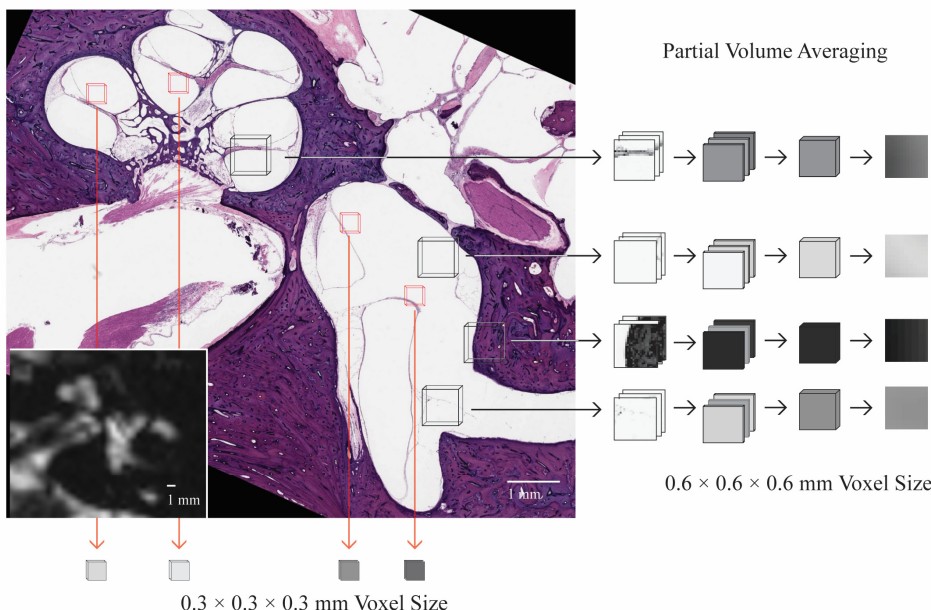

**Figure 6.** Demonstration of partial volume averaging during MR image generation. The signal from a 3-dimensional tissue volume is simplified to a 2-dimensional grayscale voxel. The shade of gray is an average of the signal from all tissues within the 3-dimensional space. Water has a high signal on CISS and is lighter. Tissue is gray, and bone and air have the lowest signal and appear dark. Smaller voxels increase the likelihood of seeing thin structures like the membranous labyrinth, which can appear blurred because of signal averaging with adjacent water.

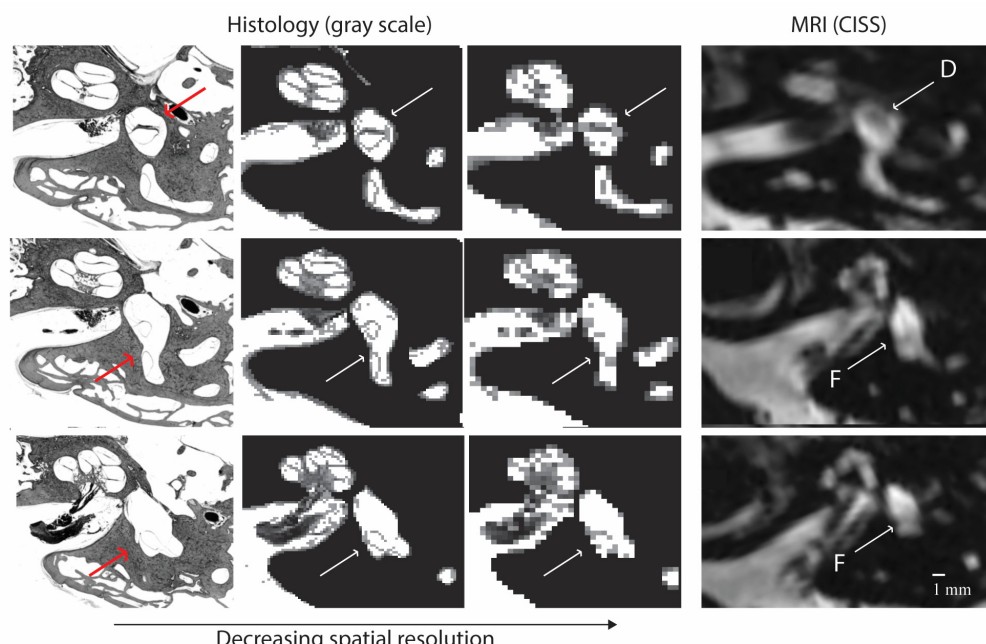

**Figure 7.** Creating a virtual MRI using histology. To show signal averaging, 2-D grayscale sections from temporal bone histopathology are down-sampled to 80 × 80 and 40 × 40-pixel grids. Bone and air were colored black to represent the expected signal from MRI, the fluid was kept white, and soft tissue of the inner ear and internal auditory canal was maintained from histology. CISS MRI is shown in similar slices. Arrows indicate the utricular macula (upper panel, object D on MRI), the anterior (middle panel, object F on MRI), and the posterior walls of the membranous utricle (lower panel, object F on MRI). Note: The example is a 2-D average. MRI averages signals in 3-D, further reducing the sharpness of the small structures when averaging [10].

CISS is a gradient echo pulse sequence in which a single radiofrequency pulse and reversing gradients are used to generate the image signal. Gradient echo sequences can be obtained more quickly than spin echo sequences. Unlike spin echo sequences, gradient echo sequences like CISS are more prone to artifacts related to inhomogeneities in the magnetic field [11,12]. While MRI magnetic fields are typically homogeneous, tissue can distort the local static magnetic field. Anatomic regions with interfaces between bone, water, and air will cause more significant distortions. These distortions commonly occur around the skull base and can cause signal loss by artificially appearing gray or black on the image and sometimes showing as a band. Banding artifacts occur in gradient echo sequences and result from off-resonance frequencies caused by these magnetic field inhomogeneities. Protons within adjacent tissue regions experiencing slightly different magnetic field strengths will not uniformly experience resonance. In sequences with rapidly alternating gradients, these regions can display as one or several bands that alternate between higher and lower signals, distorting the underlying anatomy. While banding artifacts can occur in the ear (Figure 8A,B), these were uncommonly encountered in this study and, when present, tended to be seen in numerous areas of the scan. However, banding artifacts of the eye were seen commonly in this study. Banding artifacts were identified in 96.7% of the right and 93.3% of the left eye, with an average of 1.47 bands on the right and 1.27 on the left in our cohort. The number and direction of the bands varied across participants (Figure 8C–E). This might suggest biological processes of the eye interacting with the magnetic field, causing changes in the direction of the bands.

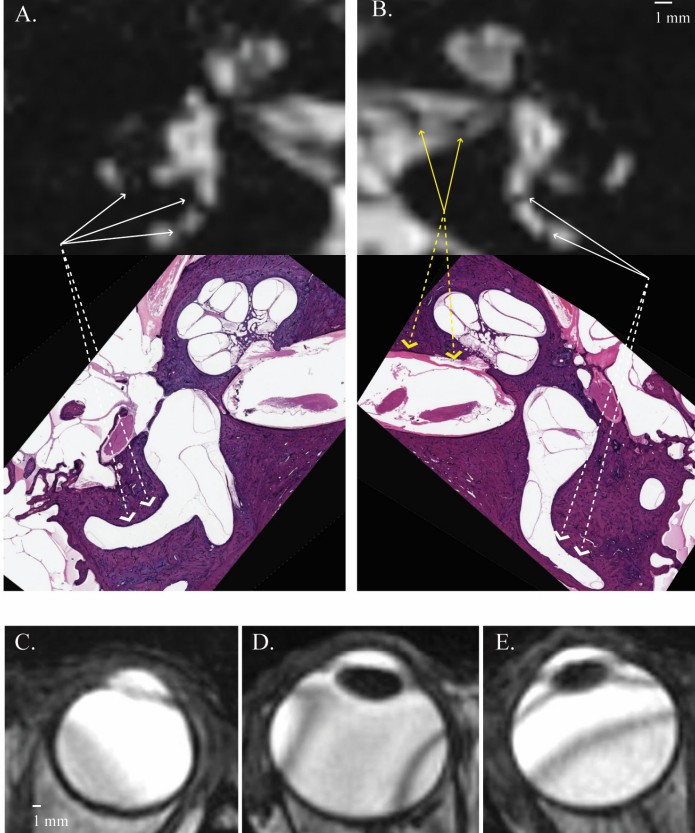

**Figure 8.** Axial MRI CISS sequence image showing a rare example of banding artifacts in the right (**A**) and left (**B**) ears (white arrows) of a single subject that showed banding artifacts in many locations, including the internal auditory canal ((**B**), yellow arrows), and the more common finding of banding artifacts in the eye, with variable orientations (**C–E**).

Attempts have been made to reduce banding artifacts by optimizing imaging parameters, such as combining multiple phase cycling angles or dynamically changing radial

acquisition of the image during a single shot [12,13]. Balanced Steady-State Free Precession (bSSFP) techniques can capture images efficiently with excellent contrast but are similarly vulnerable to an inhomogeneous magnetic field [14]. The CISS and the FIESTA sequences modify the bSSFP techniques, which reduce phase shift errors that can lead to these banding artifacts [14]. Additionally, CISS sequences, as used in our study, reduce CSF pulsation artifacts due to intrinsic flow compensation [14]. Multiple model-based approaches, such as the fast linear algorithm LORE or the introduction of the elliptical signal model, have been proposed to diminish banding artifacts [13,15]. Another attempt used nonlinear averaging (NLA) to reduce banding while maintaining high signal-to-noise, yielding high-resolution inner ear images [16]. While CISS improves over prior sequences, this study supports that banding artifacts can still occur on CISS sequences but are not prevalent in the inner ear.

*Limitations*

This single-center study is limited by its retrospective nature. As such, images from patients undergoing scans for a skull base indication were used rather than healthy controls. Two readers from the same center detected and analyzed the regions of signal loss across the vestibule and cochlea and the banding artifacts in the eyes. Comparisons were made to the histopathology of the inner ear from a single donor, limiting the ability to directly compare the low-resolution areas of low signal in the vestibule to the inner ear of the same individual. When measuring banding artifacts in the eye, variety in the width and the direction of the bands were encountered, which led to measurement difficulties, as can be seen in Figure 8. These included a broad width occupying almost 1/3 of the area of the eyeball (Figure 8C), inconsistent artifact slopes in the magnetic field (Figure 8D), or the banding artifact overlapping with the lens (Figure 8E). Furthermore, some bands were not visible at the level of the lens and were, therefore, measured more cranially or caudally. The height adjustment could affect the consistency of the measured angles. This study exclusively involved subjects who did not have inner ear or eye diseases. Consequently, the impact of such diseases on the occurrence or characteristics of the MRI artifacts was not assessed and remains beyond the scope of our findings. Nevertheless, there is robust precedent in the literature for similar studies to help define the anatomy of previously unknown structures on imaging [17,18], including the inner ear [19,20].

**5. Conclusions**

This study identified nine regions of low signal intensity on the inner ear in T2-weighted CISS MRI sequences. Most of these low-signal areas correlated to anatomy and were inconsistent with banding artifacts. Areas of low signal intensity in the vestibule may be the partial-volume averaged walls of the membranous utricle. The banding artifacts in the eye were common, of variable angle and number, and did not correlate with pathology.

**Author Contributions:** Conceptualization, B.K.W. and B.B.; methodology, B.K.W. and A.M.; formal analysis, B.K.W., C.I.S. and A.M.; writing—original draft preparation, B.K.W. and A.M.; writing—review and editing, All authors; visualization, A.M., B.K.W. and B.B. All authors have read and agreed to the published version of the manuscript.

**Funding:** This research was funded by NIH NIDCD grant K23 DC018302 and U24 DC020850.

**Institutional Review Board Statement:** The study was conducted following the Declaration of Helsinki and approved by the Institutional Review Board of Johns Hopkins University School of Medicine (protocol number IRB00279939, first approved 26 April 2021).

**Informed Consent Statement:** Patient consent was waived due to the retrospective nature of this imaging review and the minimal risk involved in this retrospective study.

**Data Availability Statement:** Extracted data from the images can be provided upon request, but image sets will not be provided due to patient confidentiality.

**Conflicts of Interest:** The authors declare no conflict of interest.

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
