# Peer review of "Patterns of Signal Intensity in CISS MRI of the Inner Ear and Eye"

_tomography, doi:10.3390/tomography10020016_

Round 1

Reviewer 1 Report

Comments and Suggestions for Authors

The paper “Patterns of Signal Intensity in CISS MRI of the Inner Ear and Eye” by Antonia Mair reported a study that analyzed the low signal regions in CISS MRI of the inner ear and their correlation to anatomical inner ear structures, as well as the banding artifacts in eye MRIs. The topic has its meaning but it could be more interesting to the community if the study was performed on a data set including subjects with inner ear and eye diseases. The background introduction and the structure of the results and discussion could be improved. I suggest the editor to consider publication of the manuscript on Tomography upon the authors address the following questions and suggestions:

Major questions and suggestions:

1.       The introduction section lacks sufficient background and references for the readers to understand the significance of corresponding the low signal regions in MRI to anatomical structures or imaging artifacts. The reviewer suggests the authors to enrich the introduction section and emphasize the clinical importance of the aim of the study.

2.       The background of the banding artifacts in the eye MRI were not sufficiently introduced as well. Why the angles of the bands are measured? What consequences may these artifacts cause?

3.       Although this study was performed on skull MRIs that includes both the inner ear and eye regions. The analysis for the two subjects – the inner ear, where the authors are correlating low intensity areas to anatomical structures, and the eye, where the authors are studying the existence of banding artifacts and the features of the artifacts – are still relatively two separate topics. The reviewer agrees that including them in a paper is in line of the readers’ interest but suggests that the authors revise the structure of the manuscript to present the two analysis clearer, for example introducing subtitles in the results and discussion section.

4.       What is the clinical interest of analyzing the banding artifacts of eye MRI?

5.       The study was performed on healthy subjects. It might be more of the interest to the readers that the authors include images that from patients who has inner ear and eye diseases, especially tumors, where the MRI images could be more complicated, and the banding artifacts can be more harmful to making diagnosis. Is this something the authors consider doing in the future or what is the difficulty of doing this?

Minor questions and suggestions:

1.       Please add scale bars to the figures.

Author Response

We want to thank the reviewers for their thoughtful reviews.  We have revised the manuscript to address their concerns and have responded to them on a point-by-point basis below. 

Reviewer 1 -

The paper “Patterns of Signal Intensity in CISS MRI of the Inner Ear and Eye” by Antonia Mair reported a study that analyzed the low signal regions in CISS MRI of the inner ear and their correlation to anatomical inner ear structures, as well as the banding artifacts in eye MRIs. The topic has its meaning but it could be more interesting to the community if the study was performed on a data set including subjects with inner ear and eye diseases. The background introduction and the structure of the results and discussion could be improved. I suggest the editor to consider publication of the manuscript on Tomography upon the authors address the following questions and suggestions:

Thanks very much for the suggestion to perform a study on patients with inner ear diseases.  While this interests us for the future, we wanted first to establish what was present in cases without inner ear disease. This was important to understand better what an artifact might be rather than a pathological process. 

Major questions and suggestions:

  1. The introduction section lacks sufficient background and references for the readers to understand the significance of corresponding the low signal regions in MRI to anatomical structures or imaging artifacts. The reviewer suggests the authors to enrich the introduction section and emphasize the clinical importance of the aim of the study.

Radiologists often look for changes in signal intensity to identify pathological processes.  The inner ear signal changes are often dismissed as related to artifacts, but this has not been explored.  We expand on this now in the introduction. 

The last sentences in the introduction were edited to state the following:

Accurately distinguishing between artifacts and anatomical or pathological structures is essential in MRI interpretation to ensure accurate diagnosis and prevent unnecessary further investigations. This study aimed to analyze banding artifacts on axial MRI CISS sequences of the inner ear and eye, addressing a critical gap in the literature on interpreting MRI of the inner ear.”

  1. The background of the banding artifacts in the eye MRI were not sufficiently introduced as well. Why the angles of the bands are measured? What consequences may these artifacts cause?

We added the following two sentences to the introduction to clarify this:

“It has been shown that artifacts in the inferior orbital region can be mistaken for neoplastic or inflammatory orbital diseases, potentially leading to unnecessary orbital biopsies or misdiagnoses (6). To analyze if banding artifacts of the eye are potentially homogenous and thus resulting from the magnetic field or are part of biological signals underlying pathophysiological processes by diseases of the eye the angles of the bands were analyzed (7).”

  1. Although this study was performed on skull MRIs that includes both the inner ear and eye regions. The analysis for the two subjects – the inner ear, where the authors are correlating low intensity areas to anatomical structures, and the eye, where the authors are studying the existence of banding artifacts and the features of the artifacts – are still relatively two separate topics. The reviewer agrees that including them in a paper is in line of the readers’ interest but suggests that the authors revise the structure of the manuscript to present the two analysis clearer, for example introducing subtitles in the results and discussion section.

Thanks, we added title headings to distinguish the two sections of the results.  

  1. What is the clinical interest of analyzing the banding artifacts of eye MRI?

The clinical interest in analyzing the banding artifacts of eye MRI lies in their potential to obscure or mimic pathology, thus hindering accurate diagnosis and appropriate management of ocular and inner ear diseases. The inner ear and the eye share similar complex anatomical structures that can be affected by these artifacts. Recognizing the similarity and the interpretive challenges it presents has been a subject of concern in previous discussions, which led to the initiation of this study. To elucidate this concern, we have emphasized in the introduction the significant risk of misdiagnosis due to banding artifacts and have detailed how our study aims to mitigate this risk by improving MRI interpretive accuracy.

  1. The study was performed on healthy subjects. It might be more of the interest to the readers that the authors include images that from patients who has inner ear and eye diseases, especially tumors, where the MRI images could be more complicated, and the banding artifacts can be more harmful to making diagnosis. Is this something the authors consider doing in the future or what is the difficulty of doing this?

Thank you for your suggestion. Studying the impact of banding artifacts in patients with inner ear and eye diseases, particularly tumors, would add clinical depth to our findings. While we aim to extend our research to disease processes, it is essential first to establish a baseline understanding of these artifacts in healthy subjects. This foundational work is crucial for developing reliable interpretative methods that can later be applied to the variable presentations in diseased populations. Many of the diseases of the ear, such as sudden sensorineural hearing loss, Meniere’s disease, vestibular neuritis, etc., have theoretical etiologies.  As a result, when no vestibular schwannoma is identified, oftentimes, the cause of the patient’s symptoms is unknown.  There are real possibilities that MRI can enhance our ability to see inner ear pathology, but understanding the processes in inner ears without these processes first is essential.  

Minor questions and suggestions:

  1. Please add scale bars to the figures.

Scale bars were added to all the figures. 

Reviewer 2 Report

Comments and Suggestions for Authors This paper presents signal intensity patterns in CISS MRI of the inner ear and eye. It focuses to the analysis of medical images. The article is an interesting approach in terms of the use of MRI. However, in data-based methods, it is important to select them appropriately in terms of the studied artifacts. The effectiveness of the diagnosis and the thesis statement depends largely on the selection of criteria and the amount of data as well as the appropriate selection of data.

Strengths:
• The article focuses on the analysis and diagnosis of the inner ear and eye.
• A relatively precise analysis of the elements of the examined medical images was presented.
• The results of research conducted in this area were discussed.
Weaknesses:
• Lack of information about the limitations and potential errors of the algorithms in the presented diagnostics.
• Lack of information about the methodology of research conducted in this area.

Remarks:
1) I propose to describe in more detail on what basis and what results from the scope of selection of input data for the research problem.
2) The authors could refer to other teaching methods and justify the choice of the presented solution.
3) What features of the presented model are an original approach in the studied discipline?
4) There is insufficient description of research methods.
5) How does the presented research compare to other works on this topic, what is its novelty and advantage?
6) The article summary is too general.

Author Response

We want to thank the reviewers for their thoughtful reviews.  We have revised the manuscript to address their concerns and have responded to them on a point-by-point basis below.

Remarks:
1) I propose to describe in more detail on what basis and what results from the scope of selection of input data for the research problem.

Thank you for your suggestion. In response, we have included a detailed flow chart as a new figure to visually represent the selection criteria and the process we used to arrive at our dataset. This flow chart is accompanied by a narrative in the Methods section, which explains the basis of our selection and the scope of the data included in our study.

2) The authors could refer to other teaching methods and justify the choice of the presented solution.

Thank you for your suggestion. To address this, we have expanded the discussion to include comparing other established methods. We have also provided a detailed rationale for our choice of technique, emphasizing its unique ability to correlate microscopic anatomy with imaging data. It has been instrumental in enhancing our understanding of normal and pathological processes.

3) What features of the presented model are an original approach in the studied discipline?

Our model introduces an unprecedented approach to analyzing CISS of the inner ear or eye. This novelty is rooted in rigorously analyzing high-resolution scans of the inner ear and comparing this to histology, which has not been previously applied in this context. The introduction now highlights how this innovation can transform our understanding of diagnostic capabilities within MRI.

4) There is insufficient description of research methods.

Thank you for your feedback. We have enhanced the methods section to include a comprehensive description of the inclusion criteria, as requested. We are committed to ensuring that our research methods are thoroughly and transparently presented. Should any other aspects of our methodology require further clarification, we welcome your specific inquiries and will promptly address them to improve the quality and clarity of our manuscript.

5) How does the presented research compare to other works on this topic, what is its novelty and advantage?

Thanks. In response, we elaborated in the introduction and discussion sections on how our research builds upon and diverges from existing works in this field. Specifically, we have outlined the novel aspects of our approach, including the use of CISS imaging, high-resolution MRI, and the comparison to inner ear histological specimens, and discussed its advantages, such as the comparison to human tissue. This comparative analysis aims to situate our research within the broader scholarly conversation and highlight its unique contributions to the field.

6) The article summary is too general.

We appreciate your feedback regarding the generality of our article summary. We understand the importance of a concise and informative summary, especially within a 200-word limit. We tried to balance brevity with the inclusion of data. We would welcome any specific suggestions you have to refine the summary further and will make targeted adjustments to ensure it accurately reflects the core findings and significance of our research.

Reviewer 3 Report

Comments and Suggestions for Authors

This paper characterizes (banding) artifacts in the inner ear and eye, from CISS MRI images obtained from 30 patients (60 ears/eyes) obtained from Johns Hopkins Hospital between 2018 and 2022. Nine regions were identified in the MRI images and systematically examined, with the number of banding artifacts and their angle quantitatively measured (present in over 93% of eyes). The study is novel in its examination of CISS MRI banding artifacts in the globe, and may help inform future clinical use of such images.

The study is generally clearly described. However, some issues might be considered:

1. In the Materials and Methods section, exclusion conditions for the scans were given. If possible, a flowchart showing the patient/eye inclusion/exclusion numbers for each condition might be provided.

2. In the Image acquisition and post-processing subsection, it is stated that three investigators reviewed the first ten MRI scans to identify areas of low signal  intensity within the inner ear (, and) nine regions were identified and systematically assessed in the remaining (i.e. 20) scans. It might first be clarified as to how the nine regions were identified - was it by discussion after initial inspection?

3. The Statistical analyses subsection that states that "Following consensus regarding the areas of low signal intensity, the two readers each assessed ten scans (20 ears). Interrater agreement was calculated using Cohen's kappa (Ƙ), and the agreement between both investigators was evaluated via the intraclass correlation coefficient (ICC)". This appears to imply that the two readers assessed 10 different scans each, and their (independent) results were then compared. This might be clarified.

4. It is then stated that "Each investigator then reviewed half the remaining scans in 30 patients"; this might be slightly confusing, since the earlier section implies that the main analysis was performed on only the (remaining) 20 scans/patients. If possible, the distribution of scans/patients for the various analyses might be explicitly described, possibly as part of a flowchart.

5. In Figure 1, while vertex (A) is labeled, (B) does not appear to be labeled/described.

6. Similarly, in Figure 3, (A) does not appear to be labeled.

7. Since the main focus of the study is on signal loss and banding artifacts, examples of such signal loss and banding artifacts (by comparing images/corresponding regions with and without) would be illustrative, together with visual description (e.g. does signal loss correlate with poor definition, etc.)

8. In the Discussion section, MRI voxel/volume averaging is described (also in Figure 5), and it is stated that smaller voxels increase likelihood of seeing thin structures appear blurred. It might be explained as to whether this implies that larger voxels are desirable, or that larger voxels imply lower resolution (and thus more general blurring)

9. Towards the end of the Discussion section, some methods for reducing banding artifacts are introduced. It might be clarified as to whether these methods have been applied to the scans used in this study.

Author Response

We want to thank the reviewers for their thoughtful reviews.  We have revised the manuscript to address their concerns and have responded to them on a point-by-point basis below.

This paper characterizes (banding) artifacts in the inner ear and eye, from CISS MRI images obtained from 30 patients (60 ears/eyes) obtained from Johns Hopkins Hospital between 2018 and 2022. Nine regions were identified in the MRI images and systematically examined, with the number of banding artifacts and their angle quantitatively measured (present in over 93% of eyes). The study is novel in its examination of CISS MRI banding artifacts in the globe, and may help inform future clinical use of such images.

The study is generally clearly described. However, some issues might be considered:

  1. In the Materials and Methods section, exclusion conditions for the scans were given. If possible, a flowchart showing the patient/eye inclusion/exclusion numbers for each condition might be provided.

Thank you for your suggestion. In response, we have included a detailed flow chart as a new figure to visually represent the selection criteria and the process we used to arrive at our dataset. This flow chart is accompanied by a narrative in the Methods section, which explains the basis of our selection and the scope of the data included in our study.

  1. In the Image acquisition and post-processing subsection, it is stated that three investigators reviewed the first ten MRI scans to identify areas of low signal  intensity within the inner ear (, and) nine regions were identified and systematically assessed in the remaining (i.e. 20) scans. It might first be clarified as to how the nine regions were identified - was it by discussion after initial inspection?

Yes. In the Image acquisition and post-processing subsection, we clarify that the nine regions of low signal intensity within the inner ear were identified through consensus. Initially, each of the three investigators independently reviewed the first ten MRI scans. After independent assessments, we convened to discuss our findings and reconcile any discrepancies. Through this discussion, we agreed on the nine regions to be included in our further analysis.

  1. The Statistical analyses subsection that states that "Following consensus regarding the areas of low signal intensity, the two readers each assessed ten scans (20 ears). Interrater agreement was calculated using Cohen's kappa (Ƙ), and the agreement between both investigators was evaluated via the intraclass correlation coefficient (ICC)". This appears to imply that the two readers assessed 10 different scans each, and their (independent) results were then compared. This might be clarified.

 Thanks for pointing this out.  We intended to state that they reviewed the same 10 scans to determine the ICC and have clarified this. 

  1. It is then stated that "Each investigator then reviewed half the remaining scans in 30 patients"; this might be slightly confusing, since the earlier section implies that the main analysis was performed on only the (remaining) 20 scans/patients. If possible, the distribution of scans/patients for the various analyses might be explicitly described, possibly as part of a flowchart.

Good idea.  We added this to the flowchart and clarified the methods. 

  1. In Figure 1, while vertex (A) is labeled, (B) does not appear to be labeled/described.

Thanks, the figure legend was rewritten to help clarify.   

Figure 1. Measurement technique for the angle of the bands. Panel A illustrates how the vertex is marked using the lens as a reference point, and then an angle is measured between the two edges (limbs) of the band. Panel B shows the measurement of the angle at which the outer edge (limb) of the band forms with a reference plane that is established by drawing a line connecting the two optic nerves. 

  1. Similarly, in Figure 3, (A) does not appear to be labeled.

OK, this was also rewritten for clarity:

 Figure 3. Vestibular anatomy may correspond to areas of low signal intensity observed in MRI.  Line A highlights a ridge near the lateral semicircular canal, which aligns with the root of the utricle. C identifies the junction of bone just lateral to the ampulla of the lateral semicircular canal, and F demonstrates the posterior region in the vestibule. Within the vestibule, the walls of the membranous labyrinth cross the otherwise fluid-filled inner ear, which may be represented as fine, gray lines in MRI scans.

  1. Since the main focus of the study is on signal loss and banding artifacts, examples of such signal loss and banding artifacts (by comparing images/corresponding regions with and without) would be illustrative, together with visual description (e.g. does signal loss correlate with poor definition, etc.)

Banding artifacts in the ear were uncommon, and we only had a single subject that had this, as shown in Figure 7.  I added a corresponding histology image to demonstrate the lack of structures in the orientation of the banding artifacts to Figure 7. 

  1. In the Discussion section, MRI voxel/volume averaging is described (also in Figure 5), and it is stated that smaller voxels increase likelihood of seeing thin structures appear blurred. It might be explained as to whether this implies that larger voxels are desirable, or that larger voxels imply lower resolution (and thus more general blurring)

Smaller voxels increase the spatial resolution so that smaller structures that are blurred with the signals of nearby water are no longer blurred.  Larger voxels are undesirable for this purpose, i.e., lower spatial resolution.  This was now clarified in the text. 

  1. Towards the end of the Discussion section, some methods for reducing banding artifacts are introduced. It might be clarified as to whether these methods have been applied to the scans used in this study.

Thanks, yes.  The CISS protocol itself was designed as a way to reduce these artifacts from earlier versions of a similar pulse sequence.  We highlight this in the discussion now. 

Round 2

Reviewer 3 Report

Comments and Suggestions for Authors

We thank the authors for comprehensively addressing our previous comments.